# Determination of Grain Growth Kinetics of S960MC Steel

**DOI:** 10.3390/ma15238539

**Published:** 2022-11-30

**Authors:** Miloš Mičian, Martin Frátrik, Jaromír Moravec, Martin Švec

**Affiliations:** 1Faculty of Mechanical Engineering, University of Žilina, Univerzitná 8215/1, 010 26 Žilina, Slovakia; 2Faculty of Mechanical Engineering, Department of Engineering Technology, Technical University of Liberec, Studentská 1402/2, 461 17 Liberec, Czech Republic

**Keywords:** austenitic grain, grain growth kinetics, S960MC, UHSS

## Abstract

Fine-grained high-strength low-alloyed (HSLA) steels are used for their advantageous combination of mechanical properties such as high yield strength, tensile strength, ductility, and good formability. These properties are mainly based on applied grain boundary strengthening, which as the only strengthening mechanism allows for the yield strength to increase without a decrease in ductility. Therefore, any changes in grain size lead to irreversible changes in material properties. Such changes also occur during welding in the heat-affected zone (HAZ), where there is a significant change in austenitic grain. In coarse-grain HAZ, this leads to a decrease in yield strength, ductility, toughness, and fatigue strength. The paper experimentally determines the growth kinetics of austenitic grain for fine-grained HSLA steel S960MC. As a result, the values of the activation energy required for grain growth Q and the proportional constant K_0_ are determined. Knowing these values is important for numerical predictions of austenitic grain size in the HAZ. Based on these predictions, the changes in yield strength, ductility, toughness, and fatigue strength can be estimated.

## 1. Introduction

The production of steel by thermo-mechanical controlled process (TMCP) allows for the application of several strengthening mechanisms in one process. Grain boundary strengthening, dislocation strengthening, and precipitation strengthening are the most common for S960MC steel and other similar grades of high-strength steel. The reason for applying these strengthening mechanisms is to achieve high strength and ductility without increasing the carbon content and content of other alloy elements in the steel. Therefore, the steel is utilized for extremely statically and dynamically stressed constructions and is suited for cold forming. In practice, it is used for load-bearing parts of cranes, heavy machinery, and other means of transport. To improve yield strength and retain ductility, grain boundary strengthening is critical. The Hall–Petch Equation (1) can be used to express the contribution of grain boundary strengthening to the overall yield strength of material [1]:(1)$$\Delta {\sigma =k}_{y}\cdot d_{\alpha }^{-1/2}$$
where k_y_ (MPa·mm^1/2^) is a coefficient expressing the effect of ferritic grain size on the increase in yield strength and d_α_ (mm) is the mean ferritic grain size. As a result, during the manufacturing process, every effort is taken to minimize the grain size to the smallest possible size. The TMCP achieves this structure by combining controlled rolling and cooling with the precipitation of (V, Nb) (C, N)-type dispersed precipitates [2,3,4].

As it follows from the production process, the given properties of steel cannot be achieved by any of the heat treatment processes. As a result, any heat load caused by welding has an irreversible effect on the microstructure in the HAZ and degrades the original base material’s properties. These changes are reflected in a significant drop in hardness and strength at higher strength grades of TMCP steels, which are formed by non-equilibrium structural components, such as S960MC steel [5,6,7,8]. In coarse-grain HAZ (CGHAZ), the thermal load from the welding process has a significant impact on the grain size. The base material in the given area overheats in the temperature range of approximately 1000 °C to the material’s solidus temperature. During austenitization, there is an increase in austenitic grain size, the coarsing of which depends on several material constants, temperature, and soaking time. Among the other factors, grain size has a decisive influence on the decomposition of austenite during cooling. With increasing austenitic grain size, the continuous cooling transformation (CCT) diagram is shifted to longer reaction times, which increases the probability of martensite formation [9,10]. In the case of steels formed by non-equilibrium phases, such as S960MC, this fact may not be unsatisfactory. After the re-cooling, the resulting structure is significantly coarsed and has different mechanical properties compared to the original base material. Coarsed grain has the greatest effect on the decrease in strength, toughness, fatigue, and plasticity [11,12,13,14,15,16]. Therefore, it is necessary to study the grain size evolution and control the grain growth in the high temperature HAZ.

In general, it may be assumed that the reduction in grain boundary area and the ensuing release of boundary surface energy is what causes grain growth. Grain growth is primarily accomplished by moving grain boundaries rather than by merging [17,18]. This movement is then interrupted, and the movement’s direction may change. Grain can also grow within existing grain, resulting in an increase in volume [1,19].

To predict the changes in the grain size of the HAZ, equations representing the austenitic grain growth kinetics can be established. The growth kinetics of austenitic grain is described by the parabolic Equation (2) for the kinetics of isothermal grain growth, which determines grain size during isothermal heating [1,12,17,18]. In the case where the diffusion of atoms across the grain boundaries is considered to be an activated process, the constant K can be replaced by Equation (3) [18,19,20]:(2)$$D^{2}-D_{0}^{2}=K\cdot t$$
(3)$${K=K}_{0}\cdot \exp\left(-\frac{Q}{R\cdot T}\right)$$
where D is the average grain diameter, D_0_ is the initial grain diameter, t is the soaking time at a given temperature, K and K_0_ are proportional constants depending on the temperature T and the activation energy Q required for grain growth, and R is the universal gas constant.

It has been observed that the increase in mean cell size of a froth of soap bubbles is related to the parabolic growth law. This is in line with the observation that boundaries move as a result of gaseous diffusion carried on by a pressure differential proportionate to the curvature of the boundary between the two sides of a soap film. Correspondingly, in a metal, the mobility is determined by either self-diffusion or impurity diffusion, with the driving force coming from the surface tension of the curved border. Unfortunately, the parabolic equation is not obeyed by the experimental results. Because of this, the original equation has to be modified and used as the Equation (4), where exponent 2 is replaced by exponent *m*:(4)$$D^{m}-D_{0}^{m}=K\cdot t$$
where *m* expresses grain growth coefficient dependent on temperature, chemical composition, processing conditions, and impurities in alloy. Impurities in particular delay and occasionally prevent grain growth. For instance, a fraction of one percent of molybdenum or manganese in solution in iron decreases the rate of grain growth by several orders of magnitude for a given mean grain size and temperature [21,22]. Moreover, the temperature dependence of m is crucial for calculations. The K values are affected when m varies with temperature. As a result, considering a predetermined fixed m value is very prevalent; the most popular one is the value between 2 and 10 [9,12,19,20,23]. According to a literature review, the coefficient m for microalloyed low-carbon steels ranges from 5 to 10 [17,22,24].

In addition to factors that support grain growth, there are several factors that counteract the growth of austenitic grain. In most cases, these are particles (carbides and nitrides) that prevent the movement of grain boundaries. In the case of TMCP steels, precipitates of the (V, Nb) (C, N) type in particular act against grain growth [18,19,20,25,26,27]. In the literature, such anti-grain growth forces are called Zener drag forces.

Since the proportional constant K_0_ and the value of the activation energy required for grain growth Q in S960MC steel have not yet been published, the aim of this work is to determine these constants. Knowing these constants is essential for the accurate determination of grain size by computing or by numerical simulation (for example, in Sysweld software). Due to the nature of the material and significant differences in microstructure and chemical composition, it is not possible to replace these constants with those from related materials.

## 2. Materials and Methods

### 2.1. Experimental Material

Strenx 960MC steel sheet (SSAB, Stockholm, Sweden) was chosen for the experiment. The steel microstructure consists mainly of bainite, tempered martensite, and residual austenite with dispersed precipitates of V, Nb, and Ti elements [5,7,26]. The mechanical properties based on the inspection certificate are given in Table 1, and the chemical composition based on EN 10204 3.1 given by the steel manufacturer is given in Table 2. An image of the base material obtained by EBSD analysis is shown in Figure 1.

### 2.2. Determination of Activation Energy Q and Kinetic Constant of Grain Growth K

To determine the proportional constant K and activation energy Q resulting from Equations (2) and (3) with sufficient accuracy, it is necessary to perform experimental measurements. Because both values are temperature-dependent, they can be determined by the test in which they are exposed to various temperatures for varying soaking times. To determine the K and Q values of S960MC steel, samples were exposed to temperatures of 900 °C, 1000 °C, 1100 °C, and 1200 °C, with soaking times of 0.5, 1, 2, 4, 6, and 8 h at each temperature. Such a number of samples is sufficient to accurately determine the required values.

Samples measuring 3 mm × 12 mm × 50 mm were heated in a resistance furnace LAC LT 50/500/13 (LAC, Židlochovice, Czech Republic) with an argon 4.6 atmosphere to prevent surface oxidation. After removal from the furnace, the samples were immersed in water with a temperature 22 °C. The samples were then metallographically prepared for EBSD analysis on the electron microscope Tescan Mira 3 (Tescan, Brno, Czech republic) with the EBSD detector Oxford Symmetry (Oxford Instruments, Abingdon, UK). The area in the middle of the material thickness of 500 µm × 500 µm was scanned. Figure 2 shows images of samples exposed to 900 °C, 1000 °C, 1100 °C, and 1200 °C, with a soaking time 1 h.

Since the determination of the grain size of the martensitic structure by microscope software is not accurate, the mean austenitic grain size (D) was determined by a linear intercept method (according to STN EN ISO 643). The measured grain had a polyhedral shape typical for austenitic grain. The measured values are shown in Table 3.

Based on the obtained data, a graphical dependence (Figure 3) was constructed between the mean area of the austenitic grain and the soaking time for each of the temperatures. The mean area (A_mean_) was determined as the square of the mean austenitic grain size (D). A linear curve was interpolated for each temperature, from which the value of the K_T_ coefficient and the value of the mean grain size for the soaking time t = 0 s were determined (D_0_). The values of K_T_ and D_0_ are given in Table 4.

After replacing the coefficient K from Equation (2) in Equation (3), we obtain the modified Equation (5):(5)$$\frac{D^{2}-D_{0}^{2}}{t}{=K}_{0}\cdot \exp\left(\frac{-Q}{R\cdot T}\right)$$

By multiplying the logarithms of both sides of this equation, it can be argued that the logarithm of ${(D}^{2}{-D}_{0}^{2})/t$ should vary directly as the reciprocal of the absolute temperature (1/T). Therefore, the value of $\left(D^{2}{-D}_{0}^{2}\right)/t$ is equal to the given task of the slope of a grain-growth isotherm (K_T_). Figure 4 shows the natural logarithm of the slopes (K_T_) of the four lines from Figure 3 plotted as functions of the reciprocal of the absolute temperature (1/T) [18,22].

As a result, a trend line can be made, and the equation for it is stated as Equation (6).
(6)y=C·x+B

Based on the Equation (6), the slope of the line C will be used to determine the value of the activation energy required for the grain growth Q according to Equation (7). The constant B from Equation (6) will be used to calculate the value of the proportional constant K_0_ according to Equation (8).
(7)Q=−(2.3·R·C)
(8)$$K_{0}{=e}^{B}$$
where R is the universal gas constant, B and C are the constants of Equation (6), and e is the Euler number.

Based on Equations (7) and (8), the values of grain growth activation energy Q = 329.95 kJ·mol^−1^ and proportionality constant K_0_ = 2.81 × 10^−2^ mm^2^·s^−1^.

### 2.3. Preparation of Welded and Simulated Samples for Comparision Purposes

To confirm the accuracy of the calculated values of the proportional constant K_0_ and the grain growth activation energy Q, the calculated values of the grain size were compared with the real values. Real grain size values were determined from the welded joint and from the sample simulated on the Gleeble 3500 device (Dynamic Systems, Poestenkill, NY, USA). 

For comparison, the butt weld of a 3 mm thick sheet metal of S960MC was made. The welded joint was made by the GMAW method in short-circuit mode (marked as GMAW-S). Welding of the welded joint was part of the previous research published by Mičian et al. [5]. Cross-section of welded joint is shown in Figure 5. The chemical composition and mechanical properties of the base material are identical to the examined samples and are listed in Table 1 and Table 2. Welding wire (Union X96) classified as G89 5 M21 Mn4Ni2.5CrMo according to STN EN ISO 16834-A with a diameter of 1 mm was used as a filler material. Welding parameters are shown in Table 5.

The welding thermal cycle was recorded by NiCr–NiAl-type K thermocouple located in the HAZ (TC1), which was placed on the root side of the weld at a distance of 0.96 mm from the root of the weld. The thermocouple was placed in the area where the material was exposed to temperature T_max_ = 1104 °C. The heating rate was determined to be 205 °C·s^−1,^ and the cooling time t_8/5_ was 17.5 s (Figure 6). The mean grain size for a given area was determined using the linear intercept method, using the microstructure image obtained by light optical microscopy (LOM) from the center of the sample thickness using ZEISS LSM 700 device (Carl Zeiss AG, Oberkochen, Germany).

The welding procedure can be also substituted with physical simulation to achieve a comparable microstructure. For this purpose, the Gleeble 3500 device was used. The dimensions of the sample used are shown in Figure 7. A NiCr–NiAl-type K thermocouple was welded onto the sample in the middle of its length. The maximum temperature T_max_ = 1105 °C and the heating rate was determined to be 210 °C·s^−1^ and time t_8/5_ was set at 17 s based on Figure 8. The mean grain size for a given area was also determined using the linear intercept method using a microstructure image obtained from the center of the sample by LOM using ZEISS LSM 700 device.

## 3. Results and Discussion

Equations (3) and (4) were used to estimate the grain size based on the known thermal cycle. The estimate was made for a temperature of 1100 °C with a soaking time of 1 s. The temperature-dependent constant K for 1100 °C (Equation (3)) and the exponent m = 3.03 were used as an input data for calculation. Based on Equation (4), the mean grain size corresponding to a temperature of 1100 °C and soaking time of 1 s was determined to be D_calc._ = 22.0 µm.

It must be concluded that the power law (Equation (4)) is an empirical relationship, which should not be extrapolated to model grain growth for conditions outside of those for which the results have been obtained [16]. The use of the power law with m not equal to 2 is therefore empirical. The equation becomes a curve-fitting empirical expression for describing the austenite grain growth kinetics when m is utilized as a fitting parameter. However, the reported value obtained for m is widely scattered. Therefore, if given equations are used outside of the experimental conditions for which they are determined, the deviation of the results must be considered.

The microstructure image of the welded joint was taken from the thermocouple position area where the temperature 1104 °C was reached (Figure 9). The mean grain size was calculated by the linear intercept method, the value of which was determined to be D_weld._ = 20.6 µm. Based on the comparison, a 6% difference between the estimated and real grain size in the HAZ can be observed.

After the physical simulation, the microstructure image was also obtained from the sample’s center of the material thickness (Figure 10). The mean grain size was determined by the linear intercept method, the value of which was determined to be D_sim._ = 19.4 µm. The difference between the calculated value of the grain size and the grain size of the simulated sample is 12%.

The determination of exact values for S960MC steel is particularly important to achieve relevant predictions (especially in numerical simulations). It is necessary to take into account the fact that the material constants Q and K_0_ cannot be replaced by constants obtained from other materials. The reason is the significant difference between these values, which would be the cause of obtaining inaccurate results for austenitic grain size prediction in the HAZ. For example, the values of the activation energy required for grain growth Q obtained by similar experiments can range from 107 to 494 kJ.mol^−1^, depending on the material. Additionally, the value of the proportional constant K_0_ can range from 10^−5^ to 10^1^ mm^2^·s^−1^ [11,19,28,29,30]. The reason for such significant differences is the influence of several factors, such as the chemical composition or the presence and size of precipitates, which significantly affect the grain-growth kinetics.

## 4. Summary

The material constants needed to describe the kinetics of austenitic grain growth for the high-strength structural steel S960MC were experimentally determined. These constants are an essential element of the empirical equations that allow for the prediction of the grain size in the HAZ only using a known thermal cycle. According to the known grain size, changes in the yield strength, ductility, or the fatigue life of welded joints can be predicted. 

For the most accurate results of constants, the samples were exposed to temperatures of 900 °C, 1000 °C, 1100 °C, and 1200 °C for 0.5, 1, 2, 4, 6, and 8 h. The mean grain size was subsequently determined on all 24 samples using EBSD analysis. The data were graphically processed to create isotherms expressing the dependence between the mean grain size and the soaking time. Calculations based on these isotherms were used to determine the value grain growth activation energy Q = 329.95 kJ·mol^−1^ and proportionality constant K_0_ = 2.81 × 10^−2^ mm^2^·s^−1^. To verify the results, the values of calculated mean grain size were compared to real mean grain size values obtained from the welded joint and physical simulation. The comparisons showed that the values of the grain sizes measured in the HAZ of the welded joint and in the simulated sample reached values that differed from 6% to 12% compared to the values determined by the calculation for the given temperature.

## Figures and Tables

**Figure 1 materials-15-08539-f001:**
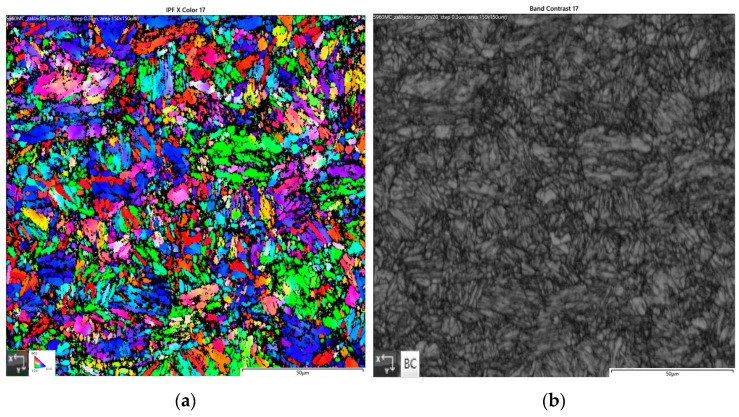
Microstructure of S960MC (EBSD): (**a**) IPF X, (**b**) band contrast.

**Figure 2 materials-15-08539-f002:**
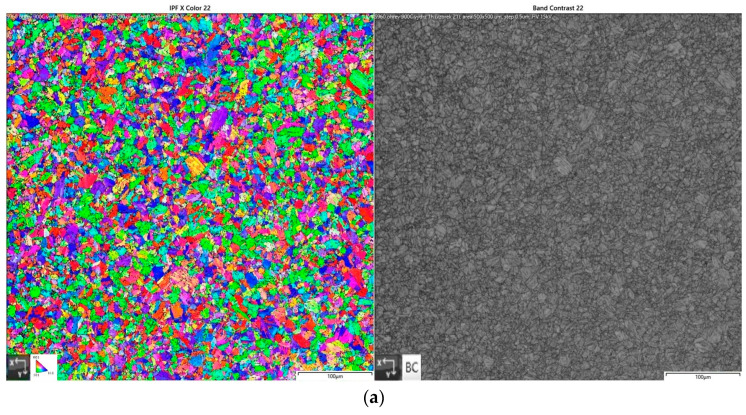
EBSD analysis of samples heated to (**a**) 900 °C, (**b**) 1000 °C, (**c**) 1100 °C, and (**d**) 1200 °C, with soaking time 1 h (IPF X—left, band contrast—right).

**Figure 3 materials-15-08539-f003:**
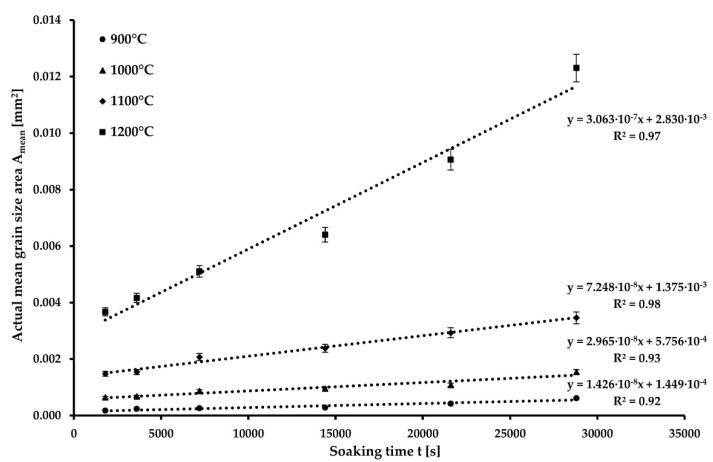
Actual mean grain area for S960MC steel vs. soaking time at evaluated temperatures.

**Figure 4 materials-15-08539-f004:**
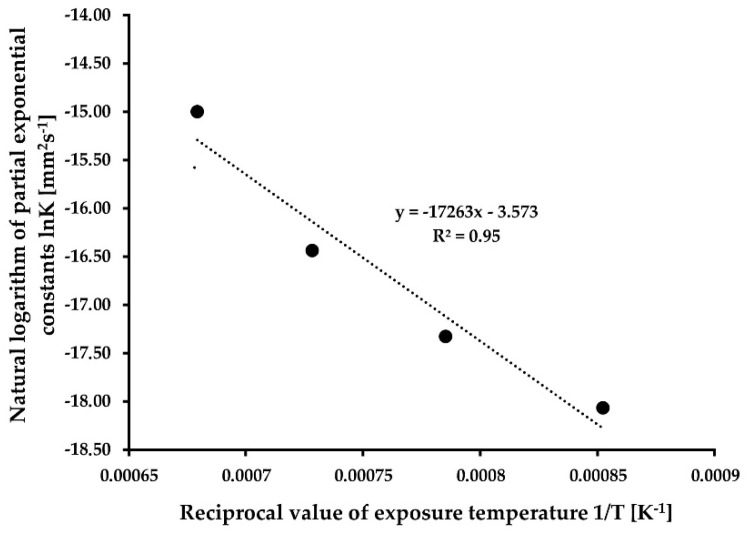
Determination of the grain growth activation energy Q and proportional constant K_0_.

**Figure 5 materials-15-08539-f005:**
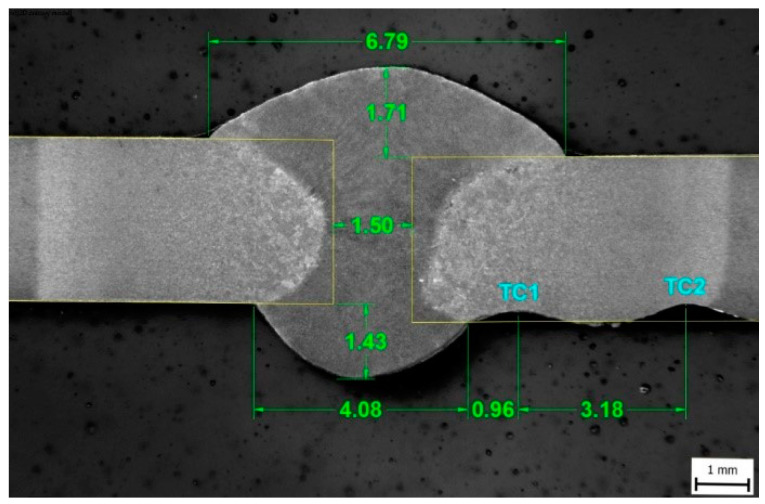
Welded joint of S960MC welded by GMAW.

**Figure 6 materials-15-08539-f006:**
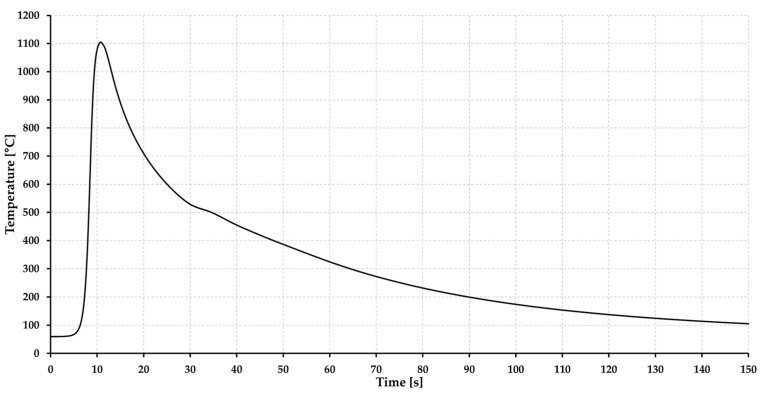
Thermal cycle of assessed welded joint [5].

**Figure 7 materials-15-08539-f007:**
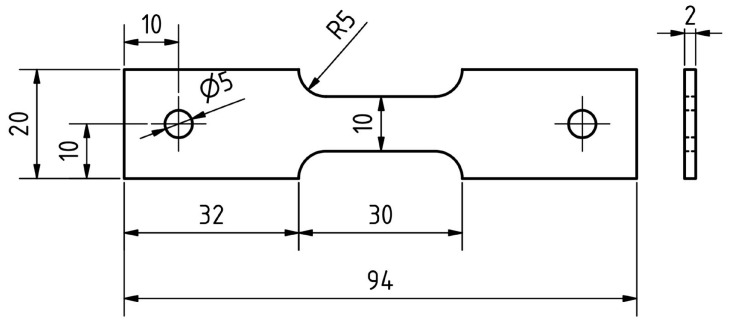
Samples for testing on a Gleeble 3500 (dimensions in mm).

**Figure 8 materials-15-08539-f008:**
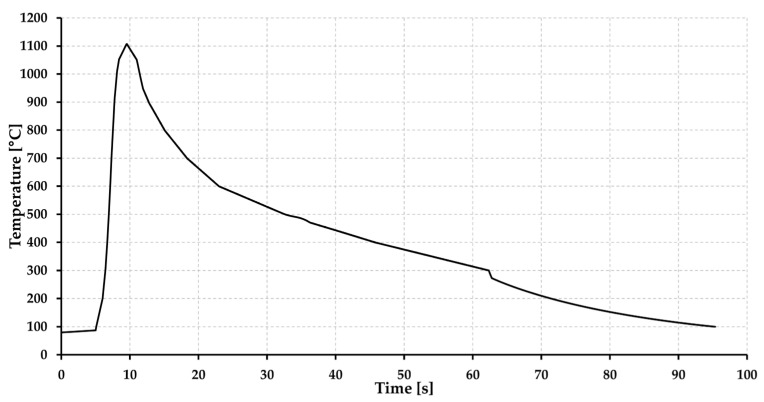
Thermal cycle of the sample simulated on Gleeble 3500 device.

**Figure 9 materials-15-08539-f009:**
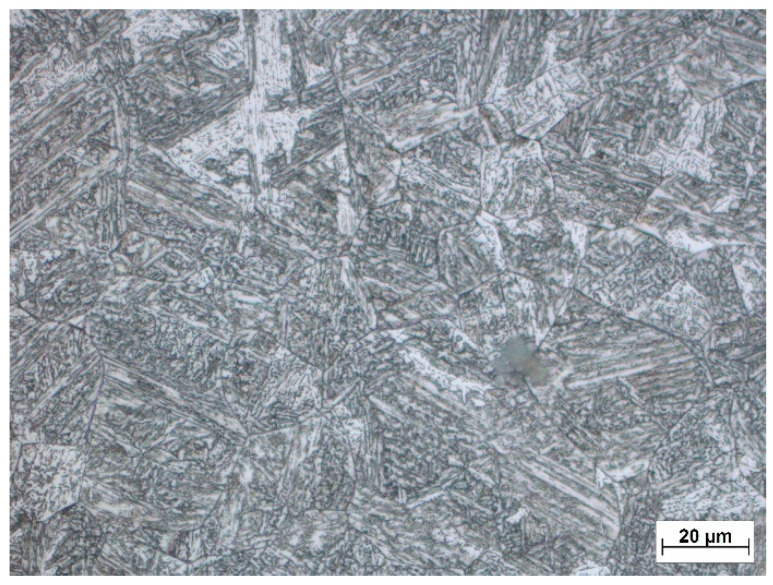
Microstructure of HAZ of welded joint exposed to temperature 1100 °C.

**Figure 10 materials-15-08539-f010:**
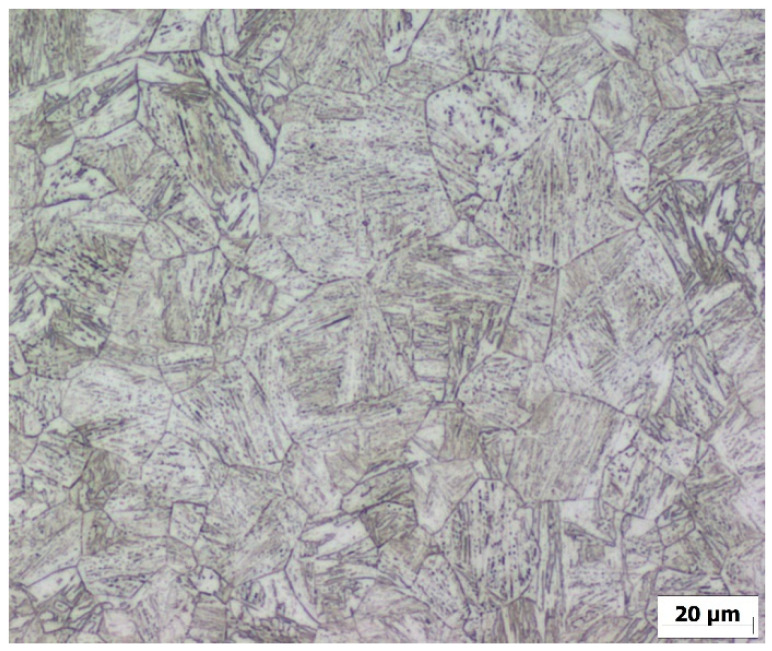
Microstructure of the sample simulated on Gleeble 3500 device exposed to temperature 1100 °C.

**Table 1 materials-15-08539-t001:** Mechanical properties of Strenx 960MC according to the inspection certificate.

R_p0.2_ (MPa)	R_m_ (MPa)	A_50mm_ (%)	CET (CEV)	KV −20 °C (J)
1018	1108	10	0.26 (0.50)	32

**Table 2 materials-15-08539-t002:** Chemical composition of Strenx 960MC according to the inspection certificate (wt.%).

**C**	**Si**	**Mn**	**P**	**S**	**Al**	**Nb**	**V**
0.085	0.18	1.06	0.01	0.003	0.036	0.002	0.007
**Ti**	**Cu**	**Cr**	**Ni**	**Mo**	**N**	**B**	**Fe**
0.026	0.01	1.08	0.07	0.109	0.005	0.0015	bal.

**Table 3 materials-15-08539-t003:** Actual mean austenitic grain size D (µm) for experimental samples of S960MC steel.

S960MC	Soaking Time (h)					
	0.5	1	2	4	6	8
**900 °C**	13.1	15.4	16.0	16.8	20.4	24.7
**1000 °C**	25.3	26.0	29.4	30.8	32.8	39.2
**1100 °C**	38.5	39.2	45.5	48.8	54.1	58.8
**1200 °C**	60.1	64.5	71.4	80.0	95.2	111.1

**Table 4 materials-15-08539-t004:** Values of initial grain size D_0_ and partial exponential constants K_T_ for a given temperature.

S960MC	K_T_ (mm^2^s^−1^)	D_0_ (mm)
**900 °C**	1.426 × 10^−8^	0.0120
**1000 °C**	2.965 × 10^−8^	0.0240
**1100 °C**	7.248 × 10^−8^	0.0371
**1200 °C**	3.063 × 10^−7^	0.0532

**Table 5 materials-15-08539-t005:** Welding parameters of welded joint [5].

Method	U (V)	I (A)	Welding Speed (cm·min^−1^)	Wire Feed Speed v_d_ (m·min^−1^)	Heat Input Q_p_ (kJ·cm^−1^)	Weld Gap b (mm)
GMAW-S	16.6	102	22.2	3.8	3.69	1.5

Heat Input value was calculated using coefficient of thermal efficiency.

## Data Availability

Not applicable.

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
