# Peer review of "Determination of Grain Growth Kinetics of S960MC Steel"

_materials, 2022, doi:10.3390/ma15238539_

Round 1
Reviewer 1 Report
Dear editors and authors, you could find my comments and suggestions in the manuscript.

Author Response
Response to Reviewer 1 Comments
Dear reviewer,
thank you very much for your valuable comments. We hope our answers and consequent article improvement will be sufficient. Please see the comments below, explaining each of your comments.
• The abbreviation did not clarify anywhere in the text. High strength low alloyed? It should be clarified.
o The abbreviation was explained in the text.
• The aim of the work was not formulated in Introduction.
o The aim of the work was formulated in Introduction section.
• This part is more suitable for Introduction (line 100-104).
o Given part was reworked and moved from Chapter 2 to Introduction section.
• The statement is not clear. There are many structures, but the authors have not identified any of them.
o You were right, based on Fig. 1, it is not possible to directly identify the described structures. Therefore, the sentence was reformulated, and references describing the structure of S960MC steel were added.
• It is known that the identification of austenitic boundaries in low-carbon steels with a martensitic structure is a really complicated issue because several packets of lath martensite can form within a certain austenite grain. Thus, there are many additional boundaries in the former austenite grain. It is quite confusing to recognize boundaries of former austenite grains in Figure 2. How did authors manage to deal with the problem?
o The austenitic grain boundaries were determined only by our estimation based on the assumed polyhedral shape of the grains. We convinced ourselves of the correctness of our estimate based on concurrently produced microstructures etched by picric acid (attached Figure). On both samples (EBSD and picric acid), approximately the same value of the mean grain size was determined by the linear method. Therefore, we considered the results obtained by the linear method from EBSD images to be relevant.
o Although, the equivalent circle diameter (ECD) values (given by the software) showed a clear trend of grain growth depending on temperature and time, they did not correspond to the actual size. Therefore, we were forced to use the linear method.
o For publication, we decided to prioritize the EBSD analysis images over the picric acid etched images (which were not published yet).
• Why did the authors use the coefficient of 2.3? It should be clarified.
o We worked based on knowledge presented in publications [18] and [19]. In a
simplified way, we took the logarithms of both sides of equations (4). After
logarithmizing the part: “exp (-Q/R∙T)” we get: “(-Q/2.3∙R∙T)”. The value 2.3 is the value corresponds to reversed value of logarithm of the Euler number (1/2.3 = 0.43 = log e). According to this relationship, it can be argued that logarithm of (D2 - D0 2)/t should vary directly as the reciprocal of the absolute temperature (1/T) and that the slope of this linear relationship is Q/(2.3 ∙ R). The value of (D2 -
D0 2)/t is equal for the given task the slope of a grain-growth isotherm (KT).
• The subtitle implies the second one. It might be better to omit the subtitle.
o It makes sense. We have removed the subtitle and at the same time we have
edited Chapter 3.

Reviewer 2 Report
The article is very well written. It presents interesting research on the growth kinetics of austenitic grain for fine-grained HSLA steel. However, unfortunately, the innovation of the results and their novelty is not well indicated, although the work is a correct analysis of the posed research problem. Therefore, I strongly suggest emphasizing the innovativeness of your work in comparison to the achievements in this field already described in the literature.
The selected material (S960MC steel) and experimental techniques are well chosen to support the posed hypothesis.
Could you please explain why you claim that the "grain size of the martensitic structure by microscope 145 software is not accurate"?
In my opinion, the uncertainty analysis should be added, especially to the values of obtained "Actual mean grain size area" (the error bars should be added in Fig.3.) and in the analysis of K0 and Q.
Could you please comment on the chosen method of grain size analysis in the context of grain shape?
Could you please explain why there is a small gap in the curve in Fig. 6 & 7 at about 28 s? Also, the curve in fig 8 is not as smooth as in Fig 6. Is it because of the chosen step? Can it be done with the smaller step? What caused the curve fault (at about 63 s)?
Minor remarks:
- p3. line 104: did you mean "other means of transport" or "different means of transport"?
- in Fig 1 &2 I would recomened to put bigger scale marker and maby more clear stereographic triangle (inverse pole figure color key)
Author Response
Response to Reviewer 2 Comments
Dear reviewer,
thank you very much for your valuable comments. We hope our answers and consequent article improvement will be sufficient. Please see the comments below, explaining each of your comments.
• I strongly suggest emphasizing the innovativeness of your work in comparison to the achievements in this field already described in the literature.
o We have added some sentences about innovativeness of our work in “Introduction” section. We have also compared our results with results of similar materials described in literature in “Results and discussion” section.
• Could you please explain why you claim that the "grain size of the martensitic structure by microscope software is not accurate"?
o Identification of austenitic boundaries in low-carbon steels with a martensitic structure is a complicated issue because several packets of lath martensite can form within a certain austenite grain. Thus, there are many additional boundaries in the former austenite grain. It is quite confusing to recognize boundaries of former austenite grains. When we wanted to use the equivalent circle diameter (ECD) values (given by the software), we showed a clear trend of grain growth depending on temperature and time, but the values did not correspond to the actual austenitic grain size. Therefore, we were forced to use the linear method.
• In my opinion, the uncertainty analysis should be added, especially to the values of obtained "Actual mean grain size area" (the error bars should be added in Fig.3.) and in the analysis of K0 and Q.
o We have made the statistical analysis and added the error bars to Fig. 3.
• Could you please comment on the chosen method of grain size analysis in the context of grain shape?
o The grain shape was commented in the text.
• Could you please explain why there is a small gap in the curve in Fig. 6 & 8 at about 28 s? Also, the curve in Fig. 8 is not as smooth as in Fig. 6. Is it because of the chosen step? Can it be done with the smaller step? What caused the curve fault (at about 63 s)
o We have checked the data from the TC and everything seems to be OK. The gap at 28 s is caused by the plotting of the curve, not by the data. Thermal cycle in Fig. 6 was measured with a frequency of 95 Hz and in Fig. 8 with a frequency of 5 Hz, which is why the differences in the graph are quite significant. Unfortunately, we cannot repeat the physical simulation at this moment with higher frequency. Also, we cannot explain the drop in the curve at around 63 s. Since our interests are primarily in the heating rate, time t8/5 and Tmax values, we have published the graph even with the curve fault at a temperature of around 300 °C.
• Minor remarks: p3. line 104: did you mean "other means of transport" or "different
means of transport"?
o Yes, it was meant as “other means of transport”.
• Minor remarks: in Fig. 1 & 2 I would recommend to put bigger scale marker and maybe
more clear stereographic triangle (inverse pole figure color key).
o We have added bigger stereographic triangle and clearer scale in Fig. 1 and 2.

Reviewer 3 Report
In this paper, the authors investigated the determines the growth kinetics of austenitic grain for fine grained HSLA steel S960MC to know kinetics parameter is substantial for numerical predictions of austenitic grain size in the HAZ. Authors are tying to perform many experiments including SEM EBSD for investigating the growth size of grain boundary as well as extensive details.
From the experiment results, it can be seen that the values of the grain sizes measured in the HAZ of the welded joint and in the simulated sample reached values that differed from 6 % to 12 % compared to the values determined by the calculation for the given temperature. The novelty of this paper is to understand of the growth kinetic parameters of grain boundary.
The whole manuscripta does not draw a clear conclusion. For example, the data in the manscripta should be summarized more clearly. And in the chapter “3 results and discussion”, there is only one section “3.1 Grain size comparison of calculated value with welded and simulated samples”. It may be some discussion sections missing.
Author Response
Response to Reviewer 3 Comments
Dear reviewer,
thank you very much for your valuable comments. We hope our answers and consequent article improvement will be sufficient. Please see the comments below, explaining each of your comments.
• The whole manuscript does not draw a clear conclusion. For example, the data in the manuscript should be summarized more clearly. And in the chapter “3 results and discussion”, there is only one section “3.1 Grain size comparison of calculated value with welded and simulated samples”. It may be some discussion sections missing.
o We have added some discussion of results and comparison with other works to the Chapter 3. We have also rewritten Chapter 4 where we put clear summary of the results.

Round 2
Reviewer 3 Report
I would be very glad to re-review the paper in greater depth. The Submission has been greatly improved and is worthy of publication.